# Efficacy Comparison between Kegel Exercises and Extracorporeal Magnetic Innervation in Treatment of Female Stress Urinary Incontinence: A Randomized Clinical Trial

**DOI:** 10.3390/medicina58121863

**Published:** 2022-12-17

**Authors:** Mislav Mikuš, Držislav Kalafatić, Adam Vrbanić, Marina Šprem Goldštajn, Mislav Herman, Marta Živković Njavro, Klara Živković, Goran Marić, Mario Ćorić

**Affiliations:** 1Department of Obstetrics and Gynecology, Clinical Hospital Center Zagreb, 10000 Zagreb, Croatia; 2School of Medicine, University of Zagreb, 10000 Zagreb, Croatia; 3Faculty of Humanities and Social Sciences, University of Zagreb, 10000 Zagreb, Croatia

**Keywords:** stress urinary incontinence, perineometry, Kegel exercises, extracorporeal magnetic innervation, randomized clinical trial

## Abstract

*Background and Objectives*: To estimate the effectiveness of Kegel exercises versus extracorporeal magnetic innervation (EMI) in the treatment of stress urinary incontinence (SUI). *Materials and Methods*: A parallel group, randomized clinical trial was conducted in the Department of Obstetrics and Gynecology, Clinical Hospital Centre Zagreb, Croatia. After assessing the inclusion/exclusion criteria, each eligible participant was randomized to one of the two observed groups by flipping a coin: the first group underwent treatment with Kegel exercises for 8 weeks, while the second group underwent EMI during the same time interval. The primary outcome was the effectiveness of treatment as measured by the ICIQ-UI-SF overall score, eight weeks after the commencement of treatment. *Results*: During the study period, 117 consecutive patients with SUI symptoms were assessed for eligibility. A total of 94 women constituted the study population, randomized into two groups: Group Kegel (*N* = 48) and Group EMI (*N* = 46). After 8 weeks of follow-up, intravaginal pressure values in the EMI group were 30.45 cmH_2_O vs. the Kegel group, whose values were 23.50 cmH_2_O (*p* = 0.001). After 3 months of follow-up, the difference was still observed between the groups (*p* = 0.001). After the end of treatment and 3 months of follow-up, the values of the ICIQ-UI SF and ICIQ-LUTSqol questionnaires in the EMI group were lower than in the Kegel group (*p* < 0.001). Treatment satisfaction was overall better in the EMI group than in the Kegel group (*p* < 0.001). *Conclusions*: Patients treated with EMI had a lower number of incontinence episodes, a better quality of life, and higher overall satisfaction with treatment than patients who performed Kegel exercises.

## 1. Introduction

Urinary incontinence (UI) is a common condition that affects the quality of life (QoL) of women at different stages of life. To date, about 50% of UI cases are due to stress urinary incontinence (SUI), which is defined as an involuntary leakage of urine during physical activity that leads to an increase in intra-abdominal pressure [1]. Despite advances in minimally invasive surgical techniques, conservative methods remain the first choice for SUI treatment [2]. Although surgical treatments show excellent results in correcting the clinical presentation of SUI and improving QoL, more than 60% of patients prefer not to undergo surgery, and nearly 15% of patients require reoperation due to disease recurrence [2]. 

The vast majority of the conservative methods available today are based on strengthening the pelvic floor muscles (PFM) to improve their response to an increase in intra-abdominal pressure [2,3,4]. According to the International Urogynecology Association (IUGA) guidelines, Kegel exercises are the first line of conservative treatment for SUI [2]. A recently published meta-analysis clearly demonstrates the benefits of Kegel exercises on the number of daily incontinence episodes and on QoL, regardless of the exercise protocol used [5]. Other conservative methods of treating SUI are intended as a safe alternative for treating patients who are uncooperative or unable to perform Kegel exercises correctly, but who wish to avoid surgical treatment or have contraindication [4]. One such method of treating SUI is extracorporeal magnetic innervation (EMI) of the PFM [6,7]. This method is based on indirect contraction of the PFM by an external magnetic field. Although there is no consensus on the optimal treatment duration of Kegel exercises and EMI, most authors suggest 6 weeks to achieve a satisfactory clinical effect [8].

Although Kegel exercises and EMI share a similar pathophysiological mechanism of action, there is currently no published randomized clinical trial (RCT) that directly compares the two methods in terms of the clinical improvement of SUI and PFM strength measured by a perineometer [9]. Therefore, the aim of this RCT is to directly compare the effectiveness of Kegel exercises versus EMI in the treatment of SUI.

## 2. Materials and Methods

### 2.1. Participants and Study Design

The detailed study design was published previously. Briefly, this parallel group RCT (ClinicalTrials.gov Identifier: NCT04307680) was conducted in the Department of Obstetrics and Gynecology, Clinical Hospital Centre Zagreb, Croatia, and was made in compliance with the SPIRIT, CONSORT and TIDieR reporting guidelines (Table 1, Figure 1). 

All enrolled participants gave written informed consent before their inclusion in the trial. The inclusion criteria were: (1) patients aged 18 to 65 years with clinically determined SUI via positive stress test; (2) patients with at least one vaginal delivery (at least 12 months ago); (3) patients with symptoms of SUI lasting at least 6 months and with a score of 6 or more on the validated ICIQ-UI SF questionnaire at baseline. Prior to treatment, a detailed gynecological examination was performed, and PFM strength was measured using a perineometer (Peritron^TM^, Laborie, Mississauga, ON, Canada). Additionally, each participant completed questionnaires assessing symptoms and QoL (ICIQ-UI SF, ICIQ-LUTSqol) and a three-day voiding diary. Each participant was then randomized to one of the two observed groups by flipping a coin (letterhead): the first group underwent treatment with Kegel exercises (letter) for 8 weeks, while the second group underwent EMI during the same time interval (head). After the completion of treatment, the study participants were again measured with a perineometer, assessed with symptom and QoL questionnaires (ICIQ-UI SF, ICIQ-LUTSqol), and the number of UI episodes was determined using a three-day voiding diary. At the next follow-up, three months after the completion of treatment, the participants’ satisfaction with the treatment was assessed using the PGI-I scale, and QoL was assessed using the ICIQ-LUTSqol questionnaire. A schematic diagram of the participant timeline is available in Figure 1.

The exclusion criteria were: pregnancy, neurological disease, positive urine cultures, previous conservative or surgical treatment of SUI, previous pelvic radiotherapy, implanted metal devices, chronic digoxin therapy, or medications that directly affect continence mechanisms.

### 2.2. Perineometer

To objectify the treatment response, we used a perineometer (Peritron^TM^, Laborie, Mississauga, ON, Canada) in both study groups. This is a noninvasive device consisting of a silicone probe that responds to changes in vaginal pressure and a central unit that records pressure changes expressed in centimeters of water. After urination, each subject is asked to contract the PFM and read the pressure resulting from the maximum voluntary contraction (MVC) of the PFM. The final value entered is the arithmetic mean of the three recorded MVCs with a rest interval of 30 s between each reading. The vaginal probe was labelled at a distance of 3.5 cm before every measurement, in order to obtain equal probe placement and to minimize measurement bias.

### 2.3. Questionnaires

Subjective instruments to assess the clinical signs and symptoms of SUI and response to treatment are based on patients’ self-assessment. ICIQ-UI SF (International Consultation on Incontinence Questionnaire—Urinary Incontinence Short Form) is one of the most commonly used self-assessment questionnaires for UI [10]. This questionnaire consists of six items, with the first two items representing demographic data and the remaining four items representing self-assessment. The score of the questionnaire ranges from a minimum of 0 to a maximum of 21 points, with a higher score indicating a worse self-assessment of incontinence. The proposed critical values categorize the results as follows: ≤5 = mild UI, 6–12 = moderate, 13–18 = severe, ≥19 = very severe UI [10]. To make the results as credible as possible, the questionnaire ICIQ-UI SF was previously validated and showed very good psychometric properties in a Croatian population [11]. 

The ICIQ-LUTSqol questionnaire (International Consultation on Incontinence Questionnaire Lower Urinary Tract Symptoms Quality of Life Module) consists of a total of 20 items and provides information about the impact of SUI on patients’ QoL, paying special attention to social aspects [12]. The self-assessment of treatment satisfaction is checked using the PGI-I scale (Patient Global Impression of Improvement) from 1 to 7, with a lower score indicating higher treatment satisfaction [6]. 

### 2.4. Interventions

#### 2.4.1. Kegel Exercises 

In the first study arm, the patients received an 8-week, high-intensity, home-based Kegel exercise program to sufficiently increase the strength, endurance, and coordination of muscle activity. The patients were initially instructed to perform 5 rapid (3-s) contractions and 10 sustained contractions (with no time limit) three times daily with 10 s relaxation periods in-between. During the first week, 200 successful contractions were targeted (1:2 ratio of rapid to sustained contractions), while in each subsequent week, the total number of contractions was increased by 10%. The patients were also instructed to use pelvic muscle contractions for urge inhibition and preventive contractions for strenuous events such as coughing, sneezing, or lifting. A description of the exercise program was emailed to the participants by a nursing assistant, both after initial enrolment and during the 8-week intervention period, which was determined by the treating physician. No additional interventions were provided to increase the effectiveness of the exercises. The patients were instructed to stop performing the exercises after the 8-week program.

#### 2.4.2. Extracorporeal Magnetic Innervation 

EMI uses strong pulsating magnetic fields to stimulate nerve activity in the pelvic floor, which in turn exercises the muscles that control bladder function, improving strength and endurance, and promoting blood flow. The treatment regimen consisted of 2 sessions per week for a total of 8 weeks. Each session lasted 30 min and consisted of a 15-min low-frequency stimulation program (10 Hz) followed by 15 min of a high-frequency stimulation program (50 Hz). The intensity of EMI was adjusted to the maximum tolerable level for the patient and was immediately discontinued if the patient reported an adverse effect. Both assigned interventions were discontinued at the participant’s request. 

#### 2.4.3. Outcomes

The primary outcome was the effectiveness of treatment, as measured by the ICIQ-UI-SF overall score eight weeks after the commencement of treatment. Secondary outcomes included:
The average increase in vaginal pressure, as measured with the Peritron^TM^ perineometer at eight weeks and 3 months after treatment.The participant’s overall treatment satisfaction, measured by the PGI-I scale three months after the end of treatment.Symptom distress, using the ICIQ-UI-SF questionnaire at the initial visit (T_0_), after the intervention was completed (T_1_), and three months (T_2_) following the end of the intervention.Quality of life, measured by ICIQ-LUTSqol also at the initial visit (T_0_), after the intervention was completed (T_1_), and three months (T_2_) following the end of the intervention.

### 2.5. Statistical Analysis

In the statistical analysis, the groups were described, and analytical statistical methods were used. Categorical variables were presented as proportions and percentages; continuous variables were presented with a median and interquartile range (25th and 75th percentile) in case of irregular distribution (e.g., duration of symptoms, parity, and vaginal delivery), or with mean and standard deviation in case of regular distribution. The regularity of the distribution of numeric variables was tested with the Shapiro-Wilk test and graphically. Then, parametric or nonparametric statistical tests were applied depending on the distribution. Differences in the distribution of the categorical variables and proportions between groups were tested with the χ2 test and, if necessary, Fisher’s exact test. Differences between the two independent groups were examined with Student’s *t* test or Mann–Whitney U test. Friedman’s analysis of variance was used to analyze repeated measures of continuous variables. To test for differences between groups, a post hoc analysis was performed using the Wilcoxon *t* test and Bonferroni correction for multiple comparisons. For the statistical analysis, IBM SPSS Statistics 25.0, Armonk, NY, USA, IBM Corp. was used, while the statistical program R was used to generate graphs. In all tests, the value *p* < 0.05 represented the level of statistical significance.

## 3. Results

### 3.1. Sample Characteristics

During the study period, 117 consecutive patients with SUI symptoms were assessed for eligibility. Among them, 16 women fulfilled at least one exclusion criteria; hence, they were excluded from the final analysis. An additional seven women were lost to follow-up. Therefore, a total of 94 women were analyzed and constituted the study population (Figure 1). The total study population was randomized into two groups: Group Kegel (*N* = 48) and Group EMI (*N* = 46). The mean age ± SD of the participants is shown in Table 1. As seen, there was no significant difference in the mean age between the groups, and no significant difference between the groups was found in the average BMI scores, nor in education level, marital status, alcohol and tobacco use, parity, previous episiotomy or caesarean section, and menopausal status (Table 2). However, in the Kegel group, there was a significantly higher proportion of retired women compared to the EMI group (20.8% vs. 2.2%, *p* = 0.012), which led to a difference in the distribution of the share of employed and unemployed in the Kegel and EMI groups (74.2% vs. 87%, and 4.2% vs. 10.9%).

### 3.2. Intravaginal Pressure Assessment with a Perineometer

The determination of intravaginal pressure with a perineometer and the comparison between the groups are shown in Table 3. Comparing the values of intravaginal pressure between the groups at baseline, there was no statistically significant difference (*p* = 0.354). After 8 weeks of follow-up, intravaginal pressure values in the EMI group were 30.45 cmH_2_O (26.2–35.2), which was statistically significantly higher compared to the Kegel group, whose values were 23.50 cmH_2_O (20.3–30.6) (*p* = 0.001). After 3 months of follow-up, the difference was still observed between the groups, i.e., the intravaginal pressure values in the EMI group were 29.15 cmH_2_O (25.3–34.2) and were statistically significantly higher compared to the Kegel group, whose values were 22.55 cmH_2_O (18.0–31.0) (*p* = 0.001).

The ANOVA for the variable intravaginal pressure in the EMI and Kegel groups are presented in Appendix A. Post hoc testing for both groups revealed a difference between the first and second measurement, and between the first and third measurement (*p* < 0.05). Moreover, there was no difference in measurements between the second and third time points in both groups (*p* > 0.05). 

### 3.3. Symptom Distress Comparison 

Determination of the value of the previously validated ICIQ-UI SF questionnaire and the comparison between the groups are shown in Table 4. Comparing the values of the ICIQ-UI SF questionnaire between the groups at baseline, there was no statistically significant difference (*p* = 0.984). After 8 weeks of follow-up, the values of the ICIQ-UI SF questionnaire in the EMI group were 8.0 points (6.0–11.0), which was statistically significantly lower compared to the Kegel group, whose values were 12.0 points (8.0–15.0) (*p* < 0.001). After 3 months of follow-up, the difference was still observed between the groups—that is, the values of the ICIQ-UI SF questionnaire in the EMI group were 7.5 points (5.0–12.0) and were statistically significantly lower compared to the Kegel group, whose values were 13.0 points (6.5–16.0) (*p* = 0.001).

The ANOVA for the variable ICIQ-UI SF in the EMI and Kegel groups are presented in Appendix A. Post hoc testing for both groups revealed a difference between the first and second measurement, and for the first and third measurement (*p* < 0.05). Moreover, there was no difference in measurements between the second and third time points in both groups (*p* > 0.05). 

### 3.4. Quality of Life Assessment

Determination of the value of the ICIQ-LUTSqol questionnaire and the comparison between the groups are shown in Table 5. At baseline, the value of the ICIQ-LUTSqol questionnaire in the total population was 60.0 points (54.0–64.0); for the EMI group it was 58.0 points (54.0–62.0); and for the Kegel group, 61.0 points (54.5–66.5). Comparing the values of the ICIQ-LUTSqol questionnaire between the groups at baseline, there was no statistically significant difference (*p* = 0.162). After the end of treatment, the values of the ICIQ-LUTSqol questionnaire in the EMI group were 39.0 points (29.0–51.0), which was statistically significantly lower compared to the Kegel group, whose values were 54.5 points (44.0–62.5) (*p* < 0.001). After 3 months of follow-up, the difference was still observed between the EMI and Kegel groups, i.e., the values of the ICIQ-LUTSqol questionnaire in the EMI group were 36.5 points (23.0–47.0) and were statistically significantly lower compared to the Kegel group, whose values were 57.5 points (44.0–64.0) (*p* < 0.001).

The ANOVA for the variable ICIQ-LUTSqol in the EMI and Kegel groups are presented in Appendix A. Post hoc testing for both groups revealed a difference between the first and second measurement, and between the first and third measurement (*p* < 0.05). Moreover, there was no difference in measurements between the second and third time points in both groups (*p* > 0.05). 

### 3.5. Voiding Diary Analysis

The voiding diary analysis and comparison between the groups is shown in Table 6. At baseline, the values of the voiding diary in the total population were 10.0 episodes/3 days (8.0–12.0)—that is, in the EMI group 10.0 (8.0–12.0) and in the Kegel group 10.5 (8.5–12.0). Comparing the values between the groups at baseline, there was no statistically significant difference (*p* = 0.616). After 8 weeks of follow-up, there was a significant decrease in incontinence episodes in the EMI group (4.0 (2.0–7.0)), which was statistically significantly lower compared to the Kegel group, whose values were 11.5 (6.0–15.0) (*p* < 0.001). After 3 months of follow-up, the difference was still observed between the groups, i.e., the voiding diary values in the EMI group were 3.0 (2.0–8.0) and were statistically significantly lower compared to the Kegel group, whose values were 10.50 (5.5–15.0) (*p* < 0.001). 

The ANOVAs for the variable voiding diary in the EMI and Kegel groups are presented in Appendix A. Post hoc testing for the EMI group revealed a difference between the first and second measurement, and between the first and third measurement (*p* < 0.05). Moreover, there was no difference in measurements between the second and third time points in the EMI group (*p* > 0.05). However, by testing the average values of the first, second, and third voiding diary results for the Kegel group, it was observed that there was no statistically significant difference between the different time points of the measurement (F = 4.719, *p* > 0.05). 

### 3.6. Overall Treatment Satisfaction 

By comparing the values of the PGI-I scale, statistically significantly higher values could be observed in the Kegel group (3.72 ± 1.31) compared to the EMI group (2.30 ± 1.01) (Appendix A).

## 4. Discussion

The results of our RCT demonstrate the superior efficacy of EMI compared to Kegel exercises in terms of the number of incontinence episodes, QoL, and perceived satisfaction with treatment. Today’s clinical work with patients suffering from SUI mainly involves conservative treatments when the clinical picture is mild to moderate, and the two most commonly used treatment modalities in Croatia and in several other countries worldwide are Kegel exercises and EMI [3,8]. The choice of treatment modality for SUI should always be based on the risk–benefit ratio and the personal preferences of the patient, and not only on the cure or improvement rate. An additional argument in favor of conservative treatment for SUI is the result of two cross-sectional surveys of patients’ opinions about the ideal approach to their treatment [6,13]. The authors pointed out that more than 80% of patients initially reject surgical treatment, primarily because of a fear of complications and the availability and simplicity of conservative treatment options.

Since no study has yet been conducted to directly compare these two treatment methods by objectively assessing the strength of the PFM during the study period [9], it is difficult to compare the results obtained. In early 2020, the only study that directly compared EMI and Kegel exercises in women suffering from SUI was published [14]. It was a randomized study in which 128 women with a clinically diagnosed SUI were divided into three groups: one group performed 12 treatments of EMI, the second group performed 12 Kegel exercises, and the third group was the control group. The aim of the study was to demonstrate the effects of the therapy on certain aspects of QoL, especially mental health. The authors showed a statistically significant improvement in depression symptoms and an improvement in the self-assessment of incontinence in both intervention groups compared to the control group [14]. The only difference between the group treated with EMI and the Kegel exercises was higher self-confidence in the Magnet group, as assessed by the General Self-Efficacy Scale questionnaire [14]. Despite the detailed analysis of QoL between the groups, the limitations of the Polish authors’ research are evident. Both groups were treated for two weeks less than expected (6 weeks instead of 8 weeks), and there was no follow-up period for the patients and no objective parameter to evaluate the treatment effect. In addition, the questionnaire used to assess the severity of incontinence was not previously validated and adapted to the Polish language, as was the case in our study of the Croatian population. Until now, EMI has mostly been compared with a placebo or in sham-controlled trials [4,7]. These studies showed the superiority of EMI in the treatment of SUI with a low incidence of adverse effects. It should be noted that there are currently no studies evaluating the pharmacoeconomic impact of specific conservative treatments for SUI, which should definitely be investigated in the future.

With the development of research in this area, new methods for quantifying PFM strength have emerged [15]. When evaluating the strength of the PFM in women, the MVC is the result of the isometric contraction produced by the voluntary contraction of the PFM [16]. We can detect this phenomenon during a clinical inspection. However, perineum retraction only confirms the presence of contraction and does not determine muscle strength. In addition to subjective assessment by inspection and palpation (using the so-called Oxford scale of PFM strength), the indirect assessment of PFM strength is possible using a perineometer [16,17]. Several studies have shown good to excellent reliability with repeated measurements, with a correlation coefficient for the MVC parameter ranging from 0.88 to 0.97 [17]. Although there is no consensus on the number of PFM contractions required to assess strength and monitor treatment, three contractions with sufficient rest time are commonly used for assessment. This approach can be explained by the fact that women often need more than one attempt to contract the PFM properly. The method of analyzing the data obtained in this way is controversial, where the average value of the three contractions or only the value of the strongest contraction should be included in the final analysis. We chose an approach that involves analysis of the mean of three contractions with sufficient rest time between each contraction, thus, reducing the possibility of over- or underestimation of the data collected and making the analysis more representative. The advantages of the perineometer as a diagnostic tool are its relatively low price, feasibility, noninvasiveness, and safety of use. However, the major disadvantage of perineometry is the difficulty in reproducing results, as at least five different perineometer models are currently on the market [17]. Although palpation and perineometry are currently the most commonly used methods for assessing PFM strength, there is currently no gold standard for quantifying PFM strength [15].

Regarding the study population, the SUI diagnosis was based on the clinical assessment. Reflecting on the diagnostic procedures used for SUI diagnosis, we believe that an invasive diagnostic approach (e.g., urodynamic evaluation) is avoidable. Although some authors argue that it can provide depth to the understanding of overall SUI pathophysiology, urodynamic evaluation is invasive, relatively expensive, and does not affect clinical decision making. In a study by Norton et al., it was estimated that in women with uncomplicated SUI, up to 33 million US dollars could be saved annually by not performing urodynamic evaluation [18].

The strengths of this study are in its design, the combination of subjective and objective parameters in the symptom distress evaluation, and the questionnaire validation before its implementation on Croatian patients. The limitation of the study is that we recruited patients from a tertiary referral urogynecological center. In such a setting, we assume to have more patients with worse SUI symptomatology; therefore, the result generalization is limited.

## 5. Conclusions

In conclusion, patients treated with EMI had a lower number of UI episodes, better QoL, and higher overall satisfaction with treatment than patients who performed Kegel exercises during the same period, as confirmed by an analysis of data from a three-day bladder diary and a specific questionnaire on QoL and treatment satisfaction. The comparison of intravaginal pressure values with a perineometer between the groups showed a greater relative increase in PFM strength in the group treated with EMI compared with the group performing Kegel exercises. Although perineometry may be useful as a new method when attempting to objectify the clinical picture before and after treatment, diagnostic tools based on patient assessment should not be disregarded. As there are currently no absolute reference values for intravaginal pressure and because PFM strength is an individual characteristic of each patient, perineometry cannot be used in the current identification of candidates for surgical treatment.

## Figures and Tables

**Figure 1 medicina-58-01863-f001:**
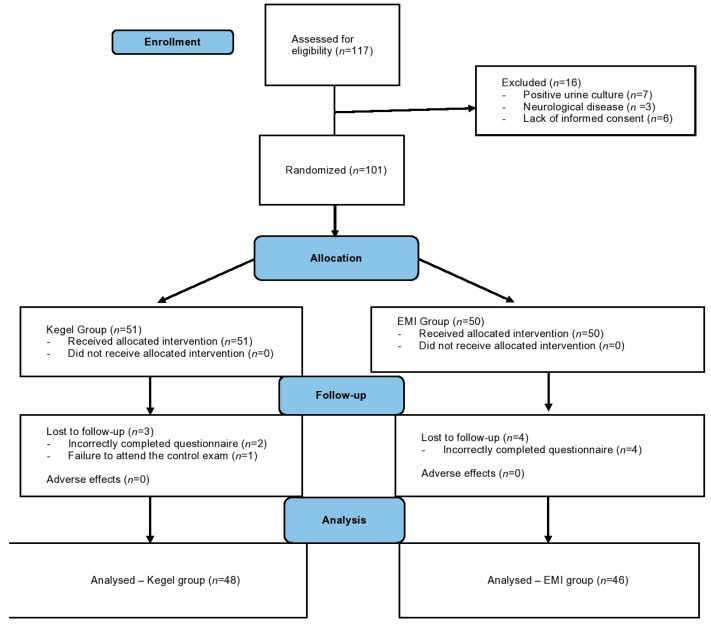
Study flow diagram according to CONSORT guidelines. Note: EMI group = extracorporeal magnetic innervation group.

**Table 1 medicina-58-01863-t001:** Time schedule of enrolment, interventions, assessments, and visits for participants.

TIMEPOINT	Pre-Study Screening, Enrolment and Consent	Baseline/Randomisation and Allocation	t_0_ Prior to Treatment	t_1_ after Treatment	t_2_ 3 Months after End of Treatment
ENROLMENT: Eligibility screen Informed consent Allocation					
	X				
	X				
		X			
INTERVENTIONS: Kegel exercises Extracorporeal magnetic innervation (EMI)					
			8 weeks		
			8 weeks		
ASSESSMENTS: Baseline variables Primary outcome variables Secondary outcome variables PGI-I scale assessment					
	X	X			
			X	X	X
			X	X	X
					X

**Table 2 medicina-58-01863-t002:** Anthropometric and demographic data of the study population.

	Total (*n* = 94)	EMI Group (*n* = 46)	Kegel Group (*n* = 48)	*p*
Age, years	48.33 ± 7.5	47.45 ± 7.4	49.16 ± 7.6	0.272
Height, cm	166.29 ± 6.9	167.21 ± 6.9	165.41 ± 6.9	0.211
Weight, kg	71.00 (62.0–81.0)	69.00 (61.0–81.0)	73.00 (62.7–80.5)	0.612
BMI, kg/m^2^	26.30 ± 5.6	25.74 ± 5.46	26.83 ± 5.7	0.348
Tobacco use, *n* (%)	48 (51.1)	27 (58.7)	21 (43.7)	0.214
Alcohol use, *n* (%)	8 (8.5)	3 (6.2)	5 (10.9)	0.665
Menopause, *n* (%)	39 (41.5)	19 (41.3)	20 (41.7)	0.826
Hysterectomy, *n* (%)	12 (12.8)	8 (17.4)	4 (8.3)	0.314
Duration of symptoms, months	12.00 (12.0–24.0)	12.00 (12.0–24.0)	15.00 (12.0–27.0)	0.373
Childbirth, *n* (%)	92 (97.9)	45 (97.8)	47 (97.9)	0.999
Parity, *n*	2.0 (2.0–3.0)	2.0 (2.0–3.0)	2.0 (1.75–3.0)	0.128
Vaginal delivery, *n*	2.0 (2.0–3.0)	2.0 (2.0–3.0)	2.0 (1.0–3.0)	0.282
Episiotomy, *n* (%)	61 (66.3)	30 (66.7)	31 (66.0)	0.882
Vacuum/forceps use, *n* (%)	11 (12.0)	8 (17.8)	3 (6.4)	0.096
Cesarean section, *n* (%)	11 (11.9)	5 (11.1)	6 (12.8)	0.806
Newborn >4000 g, *n* (%)	37 (40.2)	19 (42.2)	18 (38.3)	0.701
Marital status, *n* (%)				0.174
Married	78 (83.0)	37 (80.4)	41 (85.4)	
Unmarried	2 (2.1)	2 (4.3)	0 (0)	
Divorced	9 (9.6)	6 (13.0)	3 (6.2)	
Widow	1 (2.2)	1 (2.2)	4 (8.3)	
Employment, *n* (%)				0.012
Employed	76 (80.9)	40 (87.0)	36 (75.0)	
Unemployed	7 (7.4)	5 (10.9)	2 (4.2)	
Retired	11 (11.7)	1 (2.2)	10 (20.8)	
Education, *n* (%)				0.246
Elementary school	2 (2.1)	1 (2.2)	1 (2.1)	
High school	44 (46.8)	21 (45.7)	23 (47.9)	
BsC	23 (24.5)	15 (32.6)	8 (16.7)	
MA	25 (26.6)	9 (19.6)	16 (33.3)	

**Table 3 medicina-58-01863-t003:** Determination of intravaginal pressure by perineometer and comparison between EMI and Kegel groups with its respective interquartile range (25th and 75th percentile).

	Total (*n* = 92)	EMI Group (*n* = 46)	Kegel Group (*n* = 48)	*p*
Baseline, cmH_2_O	22.85 (18.8–26.6)	23.30 (19.1–27.3)	22.25 (18.1–24.6)	0.354
8 weeks, cmH_2_O	27.65 (22.6–33.8)	30.45 (26.2–35.2)	23.50 (20.3–30.6)	0.001
3 months, cmH_2_O	26.35 (21.1–33.2)	29.15 (25.3–34.2)	22.55 (18.0–31.0)	0.001

**Table 4 medicina-58-01863-t004:** Point values of the ICIQ-UI SF questionnaire with its respective interquartile range (25th and 75th percentile) in the study population and between the observed groups throughout the follow-up period.

	Total (*n* = 94)	EMI Group (*n* = 46)	Kegel Group (*n* = 48)	*p*
Baseline	14.00 (11.0–16.0)	14.00 (10.0–16.0)	14.00 (12.0–16.0)	0.984
8 weeks	10.00 (7.0–14.0)	8.00 (6.0–11.0)	12.00 (8.0–15.0)	<0.001
3 points	9.50 (6.0–14.0)	7.50 (5.0–12.0)	13.00 (6.5–16.0)	0.001

**Table 5 medicina-58-01863-t005:** Point values of the ICIQ-LUTSqol questionnaire with its respective interquartile range (25th and 75th percentile) in the study population and between the observed groups throughout the follow-up period.

	Total (*n* = 94)	EMI Group (*n* = 46)	Kegel Group (*n* = 48)	*p*
Baseline	60.00 (54.0–64.0)	58.00 (54.0–62.0)	61.00 (54.5–66.5)	0.162
8 weeks	47.50 (37.0–57.0)	39.00 (29.0–51.0)	54.50 (44.0–62.5)	<0.001
3 months	44.50 (28.0–62.0)	36.50 (23.0–47.0)	57.50 (44.0–64.0)	<0.001

**Table 6 medicina-58-01863-t006:** Analysis of the three-day bladder diary in the study population and between the observed groups throughout the follow-up period.

	Total (*n* = 94)	EMI Group (*n* = 46)	Kegel Group (*n* = 48)	*p*
Baseline	10.00 (8.0–12.0)	10.00 (8.0–12.0)	10.50 (8.5–12.0)	0.616
8 weeks	6.00 (3.0–12.0)	4.00 (2.0–7.0)	11.50 (6.0–15.0)	<0.001
3 months	7.00 (2.0–12.0)	3.00 (2.0–8.0)	10.50 (5.5–15.0)	<0.001

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
