# Peer review of "Efficacy Comparison between Kegel Exercises and Extracorporeal Magnetic Innervation in Treatment of Female Stress Urinary Incontinence: A Randomized Clinical Trial"

_medicina, 2022, doi:10.3390/medicina58121863_

Round 1

Reviewer 1 Report

Article  "Efficacy comparison between Kegel exercises and extracorporeal magnetic innervation in treatment of female stress urinary incontinence: a randomized clinical trial" with aim to estimate the effectiveness of Kegel exercises versus extracorporeal magnetic innervation (EMI) in the treatment of stress urinary incontinence (SUI) deserves to be published in the journal "Medicina". Authors have set the primary outcome the effectiveness of treatment as measured by the ICIQ-UI-SF overall score. They obtained results: intravaginal pressure values in the EMI group were 30.45 cmH2O vs. the Kegel group, whose values were 23.50 26 cmH2O. Authors concluded that patients treated with EMI had a lower number of incontinence episodes, better quality of life, and higher overall satisfaction with treatment than patients who performed Kegel exercises.    

I agree with the strength of the study as stated by the authors themselves in the discussion section; combination of subjective and objective parameters in symptom distress evaluation and questionnaire validation before its implementation on Croatian patients.

Author Response

Dear Reviewer, thank you very much for your comments made about our manuscript. We are
very grateful that you recognize the importance of this study.
On behalf of the co-authors,
Mislav Mikuš

Reviewer 2 Report

This is a well written manuscript with sound methodology and clear interpretations. The authors did efficacy comparison of two equally matched women with stress urinary incontinence. One group received extracorporeal magnetic intervention (EMI) and the other received Kegel's exercises.  They found better outcome in EMI group.

The study design is fine and strong by RCT, However,the limitations are identified correctly by authors. Quantification and type of incontinence are missing, long term follow up is missing. Are all patients with pure stress incontinence?

Author Response

Dear Reviewer, thank you very much for your comments made about our manuscript. We are very grateful that you recognize the importance of this study and that you underlined certain limitations. Longitunidal, long-term studies are very expensive in countries like Croatia, and with lower treatment adherence are difficult to finish.

Regarding the study population, the SUI diagnosis was based on the clinical assessment. Reflecting on the diagnostic procedures used for SUI diagnosis, we believe that an invasive diagnostic approach (e.g. urodynamic evaluation) is avoidable. Although some authors argue that it can provide depth to the understanding of overall SUI pathophysiology, urodynamic evaluation is invasive, relatively expensive, and does not affect clinical decision-making. In a study by Norton et al., it was estimated that in women with uncomplicated SUI, up to 33 million US dollars could be saved annually by not performing urodynamic evaluation.

We have added this part in the Discussion section.

All the changes are highlighted with “track change” function of Word in the revised version of the manuscript.

Once again, we thank the Editor and Reviewers for the precious suggestions and the opportunity to clarify the abovementioned issues. We hope that the new version of the manuscript could be considered in line with the high-quality standards of the Journal.

We remain available for any further detail you might wish to discuss.

On behalf of the co-authors,

Mislav Mikuš